# Tximeta: Reference sequence checksums for provenance identification in RNA-seq

**Michael I. Love** [1,2]*, **Charlotte Soneson** [3,4], **Peter F. Hickey** [5,6], **Lisa K. Johnson** [7], **N. Tessa Pierce** [7], **Lori Shepherd** [8], **Martin Morgan** [8], **Rob Patro** [9]

**1** Department of Biostatistics, University of North Carolina-Chapel Hill, Chapel Hill, North Carolina, United States of America, **2** Department of Genetics, University of North Carolina-Chapel Hill, Chapel Hill, North Carolina, United States of America, **3** Friedrich Miescher Institute for Biomedical Research, Basel, Switzerland, **4** SIB Swiss Institute of Bioinformatics, Basel, Switzerland, **5** Epigenetics and Development Division, The Walter and Eliza Hall Institute of Medical Research, Parkville, Victoria, Australia, **6** The Department of Medical Biology, University of Melbourne, Parkville, Victoria, Australia, **7** Department of Population Health and Reproduction, University of California, Davis, Davis, California, United States of America, **8** Roswell Park Comprehensive Cancer Center, Buffalo, New York, United States of America, **9** Department of Computer Science, University of Maryland, College Park, Maryland, United States of America

* michaelisaiahlove@gmail.com

**Data Availability Statement:** All datasets used in this manuscript are available as Bioconductor data packages used in the tximeta or fishpond package vignettes (https://bioconductor.org/packages/

## Abstract

Correct annotation metadata is critical for reproducible and accurate RNA-seq analysis. When files are shared publicly or among collaborators with incorrect or missing annotation metadata, it becomes difficult or impossible to reproduce bioinformatic analyses from raw data. It also makes it more difficult to locate the transcriptomic features, such as transcripts or genes, in their proper genomic context, which is necessary for overlapping expression data with other datasets. We provide a solution in the form of an R/Bioconductor package tximeta that performs numerous annotation and metadata gathering tasks automatically on behalf of users during the import of transcript quantification files. The correct reference transcriptome is identified via a hashed checksum stored in the quantification output, and key transcript databases are downloaded and cached locally. The computational paradigm of automatically adding annotation metadata based on reference sequence checksums can greatly facilitate genomic workflows, by helping to reduce overhead during bioinformatic analyses, preventing costly bioinformatic mistakes, and promoting computational reproducibility. The tximeta package is available at https://bioconductor.org/packages/tximeta.

## Author summary

Gene expression quantification from RNA sequencing is a common component of many research publications. In order that research findings can be computationally reproducible, it is critical that gene expression datasets are linked to the correct gene annotation, including the source of the annotation, the release number, and the location of the genes in a particular genome assembly. Often it is difficult for this critical metadata to be found for public datasets, and manually curating this information subjects the process to human error. We describe a solution for the missing metadata problem, whereby we embed a

tximeta; https://bioconductor.org/packages/fishpond), or in the case of the de novo transcriptome analysis, have been deposited to Zenodo (quantification data [Salmon output directory, tar.gz], https://doi.org/10.5281/zenodo.1486283; de novo transcriptome assembly [FASTA], https://doi.org/10.5281/zenodo.1486276; annotation file [GFF3], https://doi.org/10.5281/zenodo.2226742).

**Funding:** MIL is supported by NIH R01 HG009937, R01 MH118349, P01 CA142538, and P30 ES010126. NTP is supported by NSF PRFB 1711984. LS and MM are supported by NIH U41 HG004059. RP is supported by NIH R01 HG009937, and by NSF BIO-1564917, CCF-1750472, and CNS-1763680. The funders had no role in study design, data collection and analysis, decision to publish, or preparation of the manuscript.

**Competing interests:** I have read the journal's policy and the authors of this manuscript have the following competing interests: RP is a co-founder of Ocean Genomics.

checksum of the RNA reference sequences in the output files during the expression quantification step. Later we use this checksum for identification and automatic attachment of the correct metadata when the dataset is loaded into R for statistical analysis. We feel this paradigm of embedded checksums and subsequent metadata retrieval will also prove useful in other computational biology contexts.

This is a *PLOS Computational Biology* Software paper.

## Introduction

An RNA-seq data analysis often involves quantification of sequence read data with respect to a set of known reference transcripts. These reference transcripts may be downloaded from a database such as GENCODE, Ensembl, or RefSeq [1–3] in the form of nucleotide sequences in FASTA format and/or transcript locations in a genome in GTF/GFF (gene transfer format / general feature format). Alternatively a novel set of reference transcripts may be derived as part of the data analysis. The provenance of the reference transcripts, including their source and release number, is critical metadata with respect to the processed data. Without information about the reference provenance, computational reproducibility—re-performing the analysis with the same data and code and obtaining the same result [4]—may be difficult or impossible. Reproducibility has been set as a high-level goal for all NIH-funded research [5, 6], and so developers of bioinformatic tools should design software that promotes and facilitates computational reproducibility. Manually tracking critical pieces of metadata throughout a long-term bioinformatic project is tedious and error prone; still, manual metadata record-keeping is a common practice in RNA-seq bioinformatics. For example, a common approach to tracking the provenance of reference transcripts used during an RNA-seq quantification step would be to keep a README file in the same directory as the quantification data, with source and release information recorded.

In addition to impeding computational reproducibility, missing or wrong metadata can potentially lead to serious errors in downstream analysis: if quantification data are shared with genomic coordinates but without critical metadata about the genome version, computation of overlaps with other genomic data with mis-matching genome versions can lead to faulty inference of overlap enrichment. Additional annotation tasks, such as conversion of transcript or gene identifiers, or summarization of transcript-level data to the gene level, is made more difficult when the reference provenance is not known. Kanduri *et al.* [7] documented issues surrounding the lack of provenance metadata for BED, WIG, and GFF files, and described this problem as a "major time thief" in bioinformatics. Likewise, Simoneau and Scott [8] described information on genome assembly and annotation as "essential" for describing the computational analysis of RNA-seq data, and contended that, "no study using RNA-seq should be published without these methodological details." Simoneau and co-authors have recently performed a detailed analysis of hundreds of published RNA-seq studies, finding that the majority did not include annotation source and release information, thus hindering reproducible analysis [9].

A number of frameworks have been proposed that would solve the problem of tracking provenance in a bioinformatic analysis—provenance in the narrow sense defined above,

encompassing the source and release information of the reference sequences—as well as in a larger sense of tracking the state of all files, including data, metadata and any software used to process these files, throughout every step of an analysis. We will first review frameworks for tracking provenance of reference sequences, and secondly describe more general frameworks. The CRAM format, developed at the European Bioinformatics Institute, involves computing differences between biological sequences and a given reference so that the sequences themselves do not need to be stored in full within an alignment file [10]. Because the specific reference used for compression is critical for data integrity, CRAM includes checksums of the reference sequences as part of the file header. A partner utility called refget has been developed in order to allow for programmatic retrieval of the reference sequence from a computed checksum, which acts as an identifier of the reference sequence when reads have been aligned to chromosomes [11]. A similar approach is taken by the Global Alliance for Genomics and Health's (GA4GH) Variation Representation Specification (VR-Spec) [12], which uses a hashed checksum (or "digest") to uniquely refer to molecular variation, and by the seqrepo python package for writing and reading collections of biological sequences [13]. The NCBI Assembly database takes a different approach, by assigning unambiguous identifier strings (though not computed via a hash function) to sets of sequences comprising specific releases of a genome assembly [14]. Knowing the identifier is therefore sufficient to know the full set of sequences in the assembly.

Another approach to reduce manual metadata tracking associated with a number of reference sequences is Refgenie. Refgenie is a tool that helps with management of bundles of files associated with reference genomes, and facilitates sharing provenance information across research groups, in that the generation of resources is scripted [15]. Arkas and ARMOR are frameworks for automating bioinformatic analyses for RNA-seq, where metadata can be assembled and attached programmatically to downstream outputs [16, 17]. The pepkit framework and the basejump R package assist with organization and management of metadata in bioinformatic pipelines, though these cannot allow for *post hoc* identification of reference provenance [18, 19].

In 2015, Belhajjame *et al.* [20] introduced the concept of a "Research Object", an aggregation of data and supporting metadata produced within a specified scientific workflow. Their formulation was system-neutral, describing the requirements for production of a Research Object. The requirements touch on topics introduced above, such as the need to preserve data inputs, software versions, as well as traces of the provenance of data as it moves through the scientific workflow. Belhajjame *et al.* [20] summarized literature in the field of computational reproducibility and efforts toward extensive provenance tracking. The developers of the Common Workflow Language (CWL) [21] have defined a profile, CWLProv, for recording provenance through a workflow run, and have a number of implementations, including within cwltool [22]. The developers of CWLProv emphasized the importance of tracking versions of input data, such as reference genomes or variant databases in a scientific workflow, and they suggested to use and store stable identifiers of all data and software, as well as the workflow itself. As identifiers play such a crucial role in assuring reproducibility of workflows, the developers of CWLProv recommended the use of hashed checksums for identifiers of data, including any reference sequence, similar to the use of checksums in the CRAM format and VR-Spec, for identifying the reference or variant sequences. Gruning *et al.* [23] recommended combining systems such as Galaxy for encapsulating analysis tools with systems for tracking and capturing parameters and source data provenance to provide full computational reproducibility.

Here we describe an R/Bioconductor package, tximeta, for identification of reference transcript provenance in RNA-seq analyses via sequence checksums. It is situated among other

solutions for facilitating computational reproducibility described above, with some automation of routine tasks, such as conversion of transcript and gene names, but short of full automation of downstream statistical analyses as in Arkas and ARMOR (note, however, that ARMOR relies on tximeta for the import of transcript abundances). Tximeta captures the versions of the software packages used in import of quantification data, but does not provide full provenance tracking throughout downstream tasks as in the Research Object specification or in CWLProv. One unique aspect of tximeta in the context of RNA-seq is that our implementation can be used to identify the reference provenance *post hoc* on various shared or public datasets, regardless of whether the original analyst kept or shared accurate records of the reference transcripts that were used. Therefore it can provide some utility for bioinformatic analysts without requiring full buy-in of a particular workflow execution framework. *Post hoc* transcriptome identification is a novel functionality not offered by alternative existing pipelines for importing or creating RNA-seq count matrices in R/Bioconductor. Tximeta is similar in implementation to the CRAM format in the use of hashed checksums, but identifies the transcript sequences used during RNA-seq sample quantification rather than the genome sequence used during alignment. We see tximeta as a piece of a larger effort to create software systems that are "more amenable to reproducibility" [24].

## Design and implementation

### Indexing and quantification

Tximeta has been developed to work automatically with output from Salmon or alevin quantification tools [25, 26], although the implementation could be extended to other quantification tools that store the appropriate hashed checksum within the index and propagate this checksum to the sample output metadata. In addition, tximeta will work with any transcript-level quantification tool, as long as it is wrapped in a pipeline that writes the reference sequence checksum to a metadata file in each sample output directory. Without loss of generality, we describe the implementation referring to Salmon quantification data below. A diagram of the following workflow is shown in Fig 1.

During the indexing step, Salmon computes the hashed checksum of the cDNA sequence of the reference transcripts. The set of reference transcripts provided to Salmon will be referred to in this text as the *transcriptome*, although we note that the reference is not necessarily equal to the complete set of possible RNA transcripts in the sample. Currently, both the SHA-256 and SHA-512 [27] checksums are computed on the reference cDNA sequences alone, with transcript sequences concatenated together with the empty string (the SHA-256 checksum is currently taken as the primary identifier). Future implementations of Salmon and tximeta may use alternate hash functions for compatibility with larger efforts toward stable identifiers for sequence collections, for example, computing a hashed checksum over a lexicographically sorted set of checksums for each transcript cDNA sequence, which would provide order-invariance for the collection identifier. During quantification of a single sample, Salmon embeds the transcriptome index checksum in a metadata file associated with the sample output. For each sample, Salmon outputs a directory with a specific file structure, including files with quantification information as well as others with important metadata about the parameters. The entire directory, not just the text file with the quantification information, should be considered the output of the quantification tool.

### Import of quantification data into R

During import of quantification data into R/Bioconductor [28], leveraging the existing tximport package [29], tximeta reads the quantification data, as well as the transcriptome index

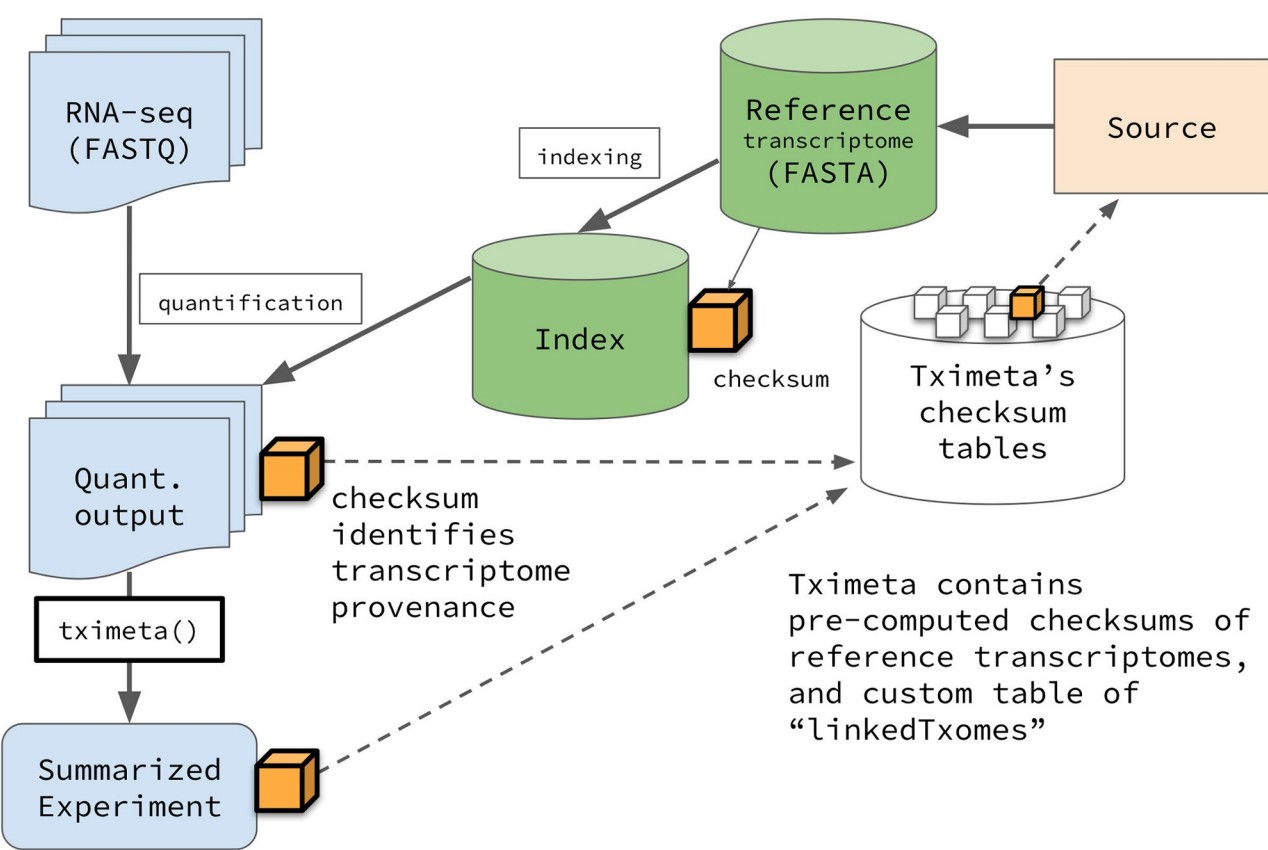

**Fig 1. Flowchart of Salmon quantification followed by tximeta.** The quantification and import pipeline results in a SummarizedExperiment object with reference transcript provenance metadata added by tximeta (see Design and Implementation). The SummarizedExperiment object contains estimated counts and other relevant metadata, and can be used with downstream statistical packages.

checksum, and compares this checksum to a hash table of pre-computed checksums of a subset of commonly used reference transcriptomes (human, mouse, and fruit fly reference transcripts from GENCODE, Ensembl, and RefSeq), as well as to a custom hash table which will be described below. Tximeta verifies that the checksum and therefore the reference transcriptome sequence is identical across all samples being imported. If there is a match of the checksum among the pre-computed checksums or in the custom hash table, tximeta will begin to compile additional relevant metadata. Depending on whether the checksum has been seen by tximeta before, one of two steps will occur:

- (First time)—Tximeta attempts to download the appropriate GTF/GFF file via FTP and parse it using Bioconductor packages. GENCODE and RefSeq GTF/GFF files are parsed by GenomicFeatures [30], while Ensembl GTF files are parsed by ensembldb [31]. Tximeta then creates a locally cached SQLite database of the parsed GTF/GFF file, as well as a GRanges object of the transcript locations [30]. The local cache is managed by the BiocFileCache Bioconductor package [32]. If the database for the correct Ensembl release is available using Bioconductor's AnnotationHub infrastructure, this pre-parsed database will be downloaded instead of downloading and parsing the GTF.

- (Subsequently)—Tximeta loads the locally cached versions of metadata (the transcript ranges, or additionally the SQLite database on demand for further annotation tasks).

After loading the appropriate annotation metadata, tximeta outputs a SummarizedExperiment object [30], a class in the Bioconductor ecosystem which stores multiple similarly shaped matrices of data, or "assays", including the estimated read counts, effective transcript lengths, and estimates of abundance (in transcripts per million, TPM). By convention, rows correspond to genomic features (e.g. transcripts or genes), while columns correspond to samples. In addition, the rows of the matrices are linked to transcript ranges, embedded in an appropriate genome version (e.g. GRCh38) including chromosome names and lengths. The SummarizedExperiment object can then be used with downstream statistical analysis packages in Bioconductor, as described in the tximeta software vignette. If tximeta did not find a matching transcriptome in the hash table then a non-ranged SummarizedExperiment will be returned as the function's output, as the location and context of the transcript ranges are not known to tximeta.

If the transcriptome was identified, and ranges were attached, then comparing data across genome versions, or without properly matching chromosomes, will produce an error, leveraging default functionality from the underlying GenomicRanges package [30]. Metadata about the samples, if provided by the user, is automatically attached to the columns of the SummarizedExperiment object. Additional metadata attached by tximeta includes all of the per-sample metadata saved from Salmon (e.g. library type, percent reads mapping, etc.), information about the reference transcriptome and file paths or FTP URLs for the source file(s) for FASTA and GTF/GFF, and the package versions for tximeta and other Bioconductor packages used during the parsing of the GTF/GFF. At any later point in time, annotation tasks can be performed by on-demand retrieval of the cached databases, for example summarization of transcript-level information to the gene level, conversion of transcript or gene identifiers, or addition of exon ranges.

A key aspect of the tximeta workflow described here is that it does not rely on self-reporting of the reference provenance for *post hoc* identification of the correct metadata. An exception to this rule is the case of a *de novo* constructed transcriptome, or in general, use of a transcriptome that is not yet contained in tximeta's built-in hash table of reference transcriptomes. For such cases, we have developed functionality in tximeta to formally link a given hashed checksum to a publicly available FASTA file(s) and a GTF/GFF file. The `makeLinkedTxome` function can be called, pointing to the transcriptome index as well as to the locations of the FASTA files and GTF/GFF file, and this will perform two operations: (1) it will add a row to a custom hash table, managed by BiocFileCache, and (2) it will produce a JSON file that can be shared or uploaded to public repositories, which links the transcriptome checksum with the source locations. When the JSON file is provided to `loadLinkedTxome` on another machine, it will add the relevant row to tximeta's custom hash table, so tximeta will then recognize and automatically populate metadata in a similar manner to if the checksum matched with a transcriptome in tximeta's built-in hash table. Finally, the cache location for tximeta, managed by BiocFileCache, can be shared across users on a cluster, for example, such that parsed databases, GRanges objects, and custom hash tables created by any one user can be leveraged by all other users in the same group.

## Comparison to related software

A number of related software projects are compared with respect to key features of tximeta in Table 1. While other RNA-seq pipelines can import quantification data into R/Bioconductor, tximeta uniquely allows for *post hoc* identification of the reference sequence provenance. The most directly related RNA-seq software packages create a SummarizedExperiment, or an object of similar shape and function, including Arkas [16], ARMOR [17], htseq [33],

**Table 1. Comparison of tximeta to related software.**

| Software | Domain | Ranges automatically attached | Release automatically attached | Post hoc lookup possible |
|---|---|:---:|:---:|:---:|
| tximeta | RNA-seq import | ✓ | ✓ | ✓ |
| tximport [29] | RNA-seq import | | | |
| Arkas* [16] | RNA-seq analysis | ✓ | ✓ | |
| ARMOR†[17] | RNA-seq analysis | ✓ | ✓ | ✓ |
| htseq [33] | RNA-seq counting | | | |
| featureCounts [34] | RNA-seq counting | ✓ | | |
| summarizeOverlaps [30] | RNA-seq counting | ✓ | ✓ | |
| pepkit [18] | Workflow management | - | - | |
| basejump [19] | Metadata utilities | - | - | |
| Refgenie [15] | Genome management | - | - | ✓ |
| CRAM+RefGet [10, 11] | Read alignment | - | - | ✓ |
| CWLProv [22] | Workflow tracing | - | - | ✓ |

Tximeta is compared to related software, grouped by domain. Columns indicate if the transcript or gene ranges are automatically attached to the output of the software, whether the transcriptome and genome release information is automatically attached, and whether *post hoc* lookup of transcriptome-related metadata is possible. A hyphen (-) indicates that the column is not directly applicable.

*Arkas attaches transcript ranges and release information for Ensembl transcripts only.

†ARMOR imports tximeta for object construction.

featureCounts [34] from the Rsubread package, and summarizeOverlaps [30] from the GenomicAlignments package. Arkas is a framework for importing transcript-level quantification data into R/Bioconductor, and specifically designed for extracting annotation metadata from Ensembl FASTA files. Arkas parses information from the FASTA header lines, and so is limited in this respect. For example, GENCODE transcript files do not contain transcript ranges in the header lines, and Ensembl header lines do not contain information about exons or their ranges. ARMOR depends on tximeta, and so relies on functionality described here to attach transcript ranges and release information to the output object.

The software packages or functions htseq, featureCounts, and summarizeOverlaps all perform counting operations for aligned RNA-seq reads with respect to specific gene models, and can be used to generate an R/Bioconductor object similar to that provided by tximeta. The htseq python package and subsequent data import with DESeq2 create a SummarizedExperiment, but without ranges or release information attached. The R functions featureCounts and summarizeOverlaps automatically attach ranges, and the latter will also attach the transcriptome release metadata, given that a GRanges object was used to perform the counting operation. However, neither featureCounts nor summarizeOverlaps allow for *post hoc* metadata operations, such as the addition or modification of ranges, or addition of relevant metadata, as they do not explicitly connect the object with a remote or locally cached database as tximeta does.

Other software such as pepkit [18], basejump [19], Refgenie [15], CRAM [10], refget [11], and CWLProv [22] are not particularly designed for RNA-seq data import, and so are less directly comparable to tximeta. Pepkit, basejump, Refgenie, and CWLProv are generic workflow or resource management tools, some of which allow for the possibility of *post hoc* identification of annotation metadata. However, none of these would provide automatic metadata attachment (range and release information) for RNA-seq data as accomplished by tximeta.

## Results

### Importing quantification data from known transcriptome

An example of importing RNA-seq quantification data using tximeta can be followed in the tximeta or fishpond Bioconductor package vignettes. Here we demonstrate the case where the Salmon files were quantified against a transcriptome that is in tximeta's pre-computed hash table. A list of supported transcriptomes as of the writing of this manuscript is provided in Table 2.

Import begins by specifying a sample table (the "column data", as the columns of the SummarizedExperiment object correspond to samples from the experiment).

```
coldata <- read.csv("coldata.csv")
```

For example, in the fishpond Bioconductor package vignette [35], the following `coldata` is read into R in the beginning of the analysis (here just showing the first two rows and five columns). The samples are from a human macrophage RNA-seq experiment [36].

```
##               names sample_id   line_id replicate condition_name
## 1 SAMEA103885102  diku_A    diku_1         1           naive
## 2 SAMEA103885347  diku_B    diku_1         1            IFNg
```

This table must have a column `files` that points to paths of quantification files (`quant.sf`), and a column `names` with the sample identifiers. The following line can be used to create the `files` column (if it does not already exist), where `dir` specifies the directory where the Salmon output directories are located, and here assuming that the sample names have been used as the Salmon output directory names.

```
coldata$files <- file.path(dir, coldata$names, "quant.sf")
```

It is expected that the quantification files are located within the original directory structure created by Salmon and with all the associated metadata files. The next step is to provide this table to the `tximeta` function, which returns a SummarizedExperiment object. If a match of the hashed checksum is found, tximeta will print a message identifying the transcriptome and will attach relevant metadata including the genomic ranges of the transcripts.

**Table 2. Pre-computed reference transcripts checksums as of early 2020.**

| Source | Organism | Releases | Transcript sequence file |
|---|---|---|---|
| GENCODE | *Homo sapiens* | 23 – 33 | transcripts.fa |
| GENCODE | *Mus musculus* | M6 – M24 | transcripts.fa |
| Ensembl | *Homo sapiens* | 76 – 99 | *.cdna.all.fa (NR) |
| Ensembl | *Mus musculus* | 76 – 99 | *.cdna.all.fa (NR) |
| Ensembl | *Drosophila melanogaster* | 79 – 99 | *.cdna.all.fa (NR) |
| Ensembl | *Homo sapiens* | 76 – 99 | *.cdna.all.fa + *.ncrna.fa |
| Ensembl | *Mus musculus* | 76 – 99 | *.cdna.all.fa + *.ncrna.fa |
| Ensembl | *Drosophila melanogaster* | 79 – 99 | *.cdna.all.fa + *.ncrna.fa |
| RefSeq | *Homo sapiens* | p1 – p12† | *_rna.fa |
| RefSeq | *Mus musculus* | p2 – p5† | *_rna.fa |

The set of pre-computed checksums span the stable releases from these sources for the years 2015—2019. (NR)—not recommended: we recommend combination of coding and non-coding transcripts for accurate RNA-seq quantification;

†—RefSeq assembly versions p13 and p6 for human and mouse respectively are currently "latest", and are subject to sequence updates under the same assembly version, and so not stable releases.

```
se <- tximeta(coldata)
```

The SummarizedExperiment object, `se`, that is returned by tximeta contains information including the estimated counts, abundances (in TPM), and the effective lengths of the transcripts. It also contains the metadata about the samples in the `colData` slot and metadata about the transcript ranges in the `rowRanges` slot. The SummarizedExperiment object can then be passed to various downstream statistical analysis packages such as DESeq2, edgeR, limma-voom, or fishpond, with example code in the tximeta software vignette [35, 37–40]. The transcript or gene ranges can be easily manipulated using the GenomicRanges or plyranges packages in the Bioconductor ecosystem [30, 41]. For example, to subset the object to only those transcripts that overlap a range defined in a variable `x`, the following line of code can be used.

```
se_sub <- se[se %over% x,]
```

The metadata columns associated with the genomic ranges of the SummarizedExperiment will have different information depending on the source. For GENCODE, Ensembl, and RefSeq, the chromosome names, start and end positions, strand, and transcript or gene ID are always included. Quantification data with an Ensembl source will also include the transcript biotype, and the start and end of the CDS sequence in the metadata columns.

Further examples of manipulating the SummarizedExperiment object can be found in the tximeta vignette, in the fishpond vignette, and in the plyrangesTximetaCaseStudy package [42].

## Importing data from a *de novo* transcriptome

It is also possible to use tximeta to import quantification data when the transcriptome does not belong to those in the set covered by pre-computed checksums (Table 2). This case may occur because the reference transcriptome is from another source or another organism than those currently in this pre-computed set, or because the transcriptome has been modified by the addition of non-reference transcripts (e.g. cancer fusion transcripts, or pathogen transcripts) which changes the checksum, or because the entire transcriptome has been assembled *de novo*. In all of these cases, tximeta provides a mechanism for local metadata linkage, as well as a formal mechanism for sharing the link between the quantification data and publicly available reference transcriptome files.

The key concept used in the case when the checksum is not part of the pre-computed set, is that of a link constructed between the transcriptome used for quantification via its hashed checksum and publicly available metadata locations (i.e. permalinks for the FASTA and GTF/GFF files). This link is created by the tximeta function `makeLinkedTxome` which stores the reference transcriptome's checksum in a custom hash table managed by BiocFileCache, along with the permalinks to publicly available FASTA and GTF/GFF files.

We demonstrate this use case with an RNA-seq experiment [43] of transcripts extracted from the speckled killifish (*Fundulus rathbuni*) quantified using Salmon [25] against a *de novo* transcriptome assembled with Trinity [44] and annotated via dammit [45]. An example workflow is provided in the denovo-tximeta repository on GitHub [46]. Here, the FASTA sequence of the *de novo* assembly as well as a GFF3 annotation file have been posted to Zenodo [47, 48], and permalinks are used to point to those records. After the reference transcripts have been indexed by Salmon, the following tximeta function can be called within R.

```
makeLinkedTxome(
  indexDir="F_rathbuni.trinity_out",
```

```
source="dammit",
organism="Fundulus rathbuni",
release="0",
genome="none",
fasta="https://zenodo.org/record/1486276/files/F_rathbuni.trinity_out.fasta",
gtf="https://zenodo.org/record/2226742/files/F_rathbuni.trinity_out.Trinity.fasta.
    dammit.gff3",
jsonFile="F_rathbuni.json"
)
```

The function does not return an R object, but has the side effect of storing an entry in the custom hash table managed by BiocFileCache, and producing a JSON file which can be shared with other analysts. The JSON file can be loaded with `loadLinkedTxome`, and it will likewise store an entry in the custom hash table of the machine where it is loaded. In either case, when the quantification data killi-quant is later imported using tximeta, the checksum will be recognized and the relevant metadata attached to the SummarizedExperiment object output. After the above function has been run, or `loadLinkedTxome` has been run, then the steps proceed as before, calling `tximeta` with an argument that specifies the sample table.

```
se <- tximeta(coldata)
```

After running `tximeta`, the SummarizedExperiment object `se` will have attached to its rows the ranges described by the GTF/GFF object, including any metadata about those transcripts. In the case of the killifish RNA-seq experiment, the transcript ranges have length, strand, and an informative column `gene_id`. The ranges of the SummarizedExperiment can be examined (here only showing the first two ranges, and suppressing range names).

```
rowRanges(se)
## GRanges object with 143492 ranges and 3 metadata columns:
##                      seqnames        ranges strand |    tx_id
##                         <Rle>     <IRanges>  <Rle> | <integer>
##   TRINITY_DN114791_c0_g1_i1        1-2308      + |      1290
##   TRINITY_DN114724_c0_g2_i1         1-635      - |      1283
##                                                      gene_id
##                                                <CharacterList>
##   ORF Transcript_...type:complete len:190 (+)
##   ORF Transcript_...5prime_partial len:83 (-)
```

## Availability and future directions

We outline an implementation for importing RNA-seq quantification data that involves (1) the quantification tool (here, Salmon) computing a hashed checksum of the reference transcript sequences, which are embedded in the index and in the per-sample output metadata, followed by (2) downstream comparison of checksums with a hash table (here, by tximeta), automated downloading and parsing of the appropriate metadata, and attachment to a rich object that bundles data and reference sequence metadata. The software is implemented within the R/Bioconductor environment for genomic data analysis, and leverages a number of existing Bioconductor packages for parsing annotation files, metadata storage, and genomic range manipulation [28, 30–32]. The tximeta package is available at https://bioconductor.org/packages/tximeta.

Currently, the pre-computed hashed checksums are focused on human, mouse, and fruit fly reference transcripts, from the popular reference transcriptome sources GENCODE, Ensembl, and RefSeq. Additional transcriptome releases from these sources are programmatically downloaded, the hashed checksum computed, and the checksum added to the tximeta

package on Bioconductor's 6 month release cycle. We are hopeful that future integration of tximeta with reference sequence retrieval efforts from the GA4GH consortium will allow for a wide expansion of the number of supported organisms. Potentially all of the releases of reference transcriptomes from Ensembl and/or RefSeq may be supported by a future reference sequence retrieval API (GENCODE releases since 2015 are already fully supported by tximeta). Furthermore, we provide a mechanism for formally linking those reference transcripts not in any pre-computed hash table (e.g. *de novo* transcriptomes) with publicly available metadata. Finally, we plan to develop tximeta to support provenance identification at the level of alleles, by combining our current reference transcript identification with transcript variant identification as described in GA4GH's Variant Representation Specification [12].

Tximeta extends tximport [29], and so is appropriate for importing transcript-level quantification data. Tximeta is not applicable to tasks such as counting of genome-aligned reads in genomic features such as exons, or ChIP- or ATAC-seq peaks. For aligned reads stored in CRAM format [10], future work along the lines of tximeta could involve programmatic utilization of genomic feature release metadata following read counting operations, for example matching exons to transcripts.

All bioinformatic software packages have limited lifespan, including the package described here. We join with others in recommending the underlying paradigm of embedding reference sequence checksums in sample output metadata, followed by downstream database lookup of checksums, and identification of reference sequence metadata. This paradigm should be adopted by other bioinformatic software that outputs any data that refers to a reference sequence. In addition, workflows can be created that wrap existing tools to ensure that hashed checksums of relevant annotation metadata are propagated to sample output directories. Such workflows have the advantage of not requiring additional effort or actions on the part of the upstream bioinformatic analyst. Otherwise, we risk exposing downstream analysts to the "major time thief" of *post hoc* guesswork involved in identifying the provenance of datasets shared publicly but without critical metadata [7].

## Acknowledgments

The authors thank the following individuals for useful discussions in the development of tximeta: Vince Carey, Paul Flicek, Joel Parker, Oliver Hofmann, Stephen Turner, Shannan Ho Sui, Thomas Keane, Andy Yates, Reece Hart, Matthew Laird, Terence Murphy, Nathan Sheffield. The authors also thank Reid Brennan, C. Titus Brown, and Andrew Whitehead for allowing use of the killifish transcriptome dataset in the *de novo* transcriptome example.

## Author Contributions

**Conceptualization:** Michael I. Love, Charlotte Soneson, Peter F. Hickey, Rob Patro.

**Data curation:** Lisa K. Johnson, N. Tessa Pierce.

**Funding acquisition:** Michael I. Love, Rob Patro.

**Methodology:** Michael I. Love, Charlotte Soneson, Peter F. Hickey, Rob Patro.

**Software:** Michael I. Love, Charlotte Soneson, Peter F. Hickey, Lori Shepherd, Martin Morgan, Rob Patro.

**Writing – original draft:** Michael I. Love.

**Writing – review & editing:** Michael I. Love, Charlotte Soneson, Peter F. Hickey, Lisa K. Johnson, N. Tessa Pierce, Lori Shepherd, Martin Morgan, Rob Patro.

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
