## [Decision Letter · Decision Letter 0]

17 Dec 2019

Dear Dr Love,

Thank you very much for submitting your manuscript 'Tximeta: reference sequence checksums for provenance identification in RNA-seq' for review by PLOS Computational Biology. Sorry for the long time it took to review your manuscript due to the difficulty in securing peer reviewers. Your manuscript has now been fully evaluated by the PLOS Computational Biology editorial team and in this case also by independent peer reviewers. The reviewers appreciated the attention to an important problem, but raised some concerns about the manuscript as it currently stands. While your manuscript cannot be accepted in its present form, we are willing to consider a revised version in which the issues raised by the reviewers have been adequately addressed. We cannot, of course, promise publication at that time.

Sincerely,

Mihaela Pertea

Software Editor

PLOS Computational Biology

Mihaela Pertea

Software Editor

PLOS Computational Biology

[LINK]

Reviewer's Responses to Questions

**Comments to the Authors:**

Reviewer #1: The manuscript "Tximeta: reference sequence checksums for provenance identification in RNA-seq" describes a software package tximeta that tracks provenance in RNA-seq analyses. The package is a drop-in replacement for a previously published tximport, where the new package tracks provenance information while performing similar operations as supported by the previous package. The idea is relatively straightforward, but potentially highly useful for improving data provenance and reproducibility of RNA-seq analyses. A desirable property is that the implementation can be used to identify the reference provenance on various shared datasets post hoc without requiring that the original analyst kept accurate records of the reference transcripts used. The relevant previous literature is comprehensively covered. It is a bit unclear how and to what degree the idea presented here brings conceptual novelty in itself, as I found the discussion of the characteristics of the proposed solution relative to existing solutions to be a bit unstructured and hard to follow. But the concept is in my opinion important and the implementation seem to operationalize this in a very useful and practical manner.

Major comments:

- As the idea itself is relatively simple, I think it should ideally be presented in a shorter form than the current nine pages. For instance, the Results section is, in my opinion, unnecessarily detailed in its present form, with excerpts and details that would seem more suited for supplementary material or a use case on a supporting web page. Examples of this include the excerpts in line 216-220 and part of the details given in line 232-237. If possible, I think it would also be advantageous to get to the point a bit quicker in the introduction and cut it down from the current two pages to maybe half. There are also aspects of the work that I currently found a bit unclear, so I would encourage to also consider my comments below in light of compactness of presentation.

- The introduction gave a very good overview of existing relevant solutions. However, I believe it could be clearer in describing what it conceptually shares with related approaches and what is unique - I found the introduction to be a bit vague in terms of describing and contrasting tximeta to the described existing solutions. A more systematic categorization of approaches and their features would probably be useful, including more explicitly providing a rationale for the current work by discussing limitations of current approaches. Also a table or figure would probably help make it even clearer. Also, is the post hoc possibilities mostly unique to this tool? As the introduction is quite long already, I would suggest to ensure that such a clarification does not increase the length.

- Furthermore, it might be useful to mention the importance and the implications of differences between solutions already in the introduction. An example is that "Tximeta is similar in implementation to the CRAM format in the use of hashed checksums, but identifies the transcript sequences used during sample quantification rather than the genome sequence used during alignment." Here, nothing is stated regarding what are the implications of such a difference.

- I would have appreciated a clearer presentation of how tximeta fits in as part of an overall reproducible analysis - what comes before and after the use of tximeta. Again, perhaps at least partly in the form of a figure.

- I would have expected to see a brief discussion of potential challenges due to the current limitation to Salmon? How much does this limit current usefulness in the author's view? Would it be problematic to have different provenance schemes for different aligners? And how does this issue relate to that the CRAM format refer to genome sequence.

Minor comments:

- The terms hash and checksum seem to be used interchangeably. Are they used in the same or slightly different meanings?

- Might be useful to separate a bit clearer what is brought by tximeta itself versus what is mostly carried over from underlying tools.

- The cover letter claim novelty, while the manuscript itself does not do so explicitly. Related to the point above regarding how it compares to existing approaches, I would encourage to discuss/argue for novelty also in the manuscript.

Reviewer #2: The authors present the tximeta Bioconductor package that provides functionality

to identify the transcriptome used during sequence alignment and automatically

download (or build) the correct annotation resource and add these annotations to

the table of quantified transcripts. This facilitates RNA-seq data analysis

considerably and helps to avoid the common pitfall of using incorrect

annotations on quantified transcript tables.

The paper is well written and the functionality of the package clearly

described. The authors provide examples for different use cases including de

novo assembled transcritomes. Also, the authors provide an extensive comparison

of their approach to other existing tools. The technically excellent package is

very well integrated into the RNA-seq data analysis workflows in Bioconductor

ensuring that it will be extensively used.

I have only some minor points the authors may consider to address

1) For cases in which alignment was performed against transcriptome sequences

from Ensembl, the pre-compiled EnsDb annotation database for the

corresponding Ensembl version which is available in Bioconductor's

AnnotationHub should be used instead of creating such a resource on-the-fly

from Ensembl's GTF/GFF3 files. Pre-build EnsDb databases are available for

all species in Ensembl and provide additional annotations such as mappings to

protein identifiers, NCBI Entrezgene identifiers or, more recent databases,

even the G-C nucleotide content of each transcript.

2) In addition to adding annotations to the transcript table, it might be useful

to have a function that returns the actual TxDb or EnsDb database from which

these annotations were taken. This would allow users to extract additional

information for the transcripts such as the number of exons of a transcript

or to even use the additional functionality of these annotation resources

such as mapping to proteins identifiers, conversion of transcriptome to

proteome coordinates.

**Have all data underlying the figures and results presented in the manuscript been provided?**

Reviewer #1: Yes

Reviewer #2: Yes

PLOS authors have the option to publish the peer review history of their article (what does this mean?). If published, this will include your full peer review and any attached files.

Reviewer #1: Yes: Geir Kjetil Sandve

Reviewer #2: Yes: Johannes Rainer

---

## [Decision Letter · Decision Letter 1]

18 Jan 2020

Dear Dr. Love,

We are pleased to inform you that your manuscript 'Tximeta: reference sequence checksums for provenance identification in RNA-seq' has been provisionally accepted for publication in PLOS Computational Biology.

Before your manuscript can be formally accepted you will need to complete some formatting changes, which you will receive in a follow up email. A member of our team will be in touch within two working days with a set of requests.

Best regards,

Mihaela Pertea

Software Editor

PLOS Computational Biology

Mihaela Pertea

Software Editor

PLOS Computational Biology

Reviewer's Responses to Questions

**Comments to the Authors:**

Reviewer #1: The authors have responded convincingly to my concerns in the previous review round, providing a comprehensive and very clear comparison to related tools and literature, as well as describing the contribution of the paper very clearly. I have no further scientific issues with the manuscript.

As I also mentioned in the previous round, my personal preference is for manuscripts to present concepts and tools as succinctly as possible so long as they deliver the main message, while I prefer that details relevant for its usage are instead presented in tutorials or other supplementary resources. I agree that the present length is quite average for a software paper in PLoS comp biol. My personal opinion is nonetheless that the particular concept (tool) presented here could have been presented more compactly without sacrificing any critical aspects, and that such a more compact form would be even more useful for readers. However, as this is not a scientific issue, I leave it to the authors and editor to decide on what they find should be included and what is then an appropriate length.

Reviewer #2: The authors have successfully addressed all my comments.

**Have all data underlying the figures and results presented in the manuscript been provided?**

Reviewer #1: Yes

Reviewer #2: Yes

PLOS authors have the option to publish the peer review history of their article (what does this mean?). If published, this will include your full peer review and any attached files.

Reviewer #1: Yes: Geir Kjetil Sandve

Reviewer #2: Yes: Johannes Rainer

---

## [Editor Report · Acceptance letter]

18 Feb 2020

PCOMPBIOL-D-19-01678R1 

Tximeta: Reference sequence checksums for provenance identification in RNA-seq

Dear Dr Love,

I am pleased to inform you that your manuscript has been formally accepted for publication in PLOS Computational Biology. Your manuscript is now with our production department and you will be notified of the publication date in due course.

With kind regards,

Sarah Hammond
